# Toxic Effects and Mechanisms of Silver and Zinc Oxide Nanoparticles on Zebrafish Embryos in Aquatic Ecosystems

**DOI:** 10.3390/nano12040717

**Published:** 2022-02-21

**Authors:** Yen-Ling Lee, Yung-Sheng Shih, Zi-Yu Chen, Fong-Yu Cheng, Jing-Yu Lu, Yuan-Hua Wu, Ying-Jan Wang

**Affiliations:** 1Department of Environmental and Occupational Health, College of Medicine, National Cheng Kung University, Tainan 70428, Taiwan; yenpig8291@gmail.com (Y.-L.L.); a08916@yahoo.com.tw (Y.-S.S.); q781001@gmail.com (Z.-Y.C.); annie880308@gmail.com (J.-Y.L.); 2Department of Oncology, Tainan Hospital, Ministry of Health and Welfare, Tainan 70101, Taiwan; 3Department of Chemistry, Chinese Culture University, Taipei 11114, Taiwan; zfy3@ulive.pccu.edu.tw; 4Department of Oncology, National Cheng Kung University Hospital, College of Medicine, National Cheng Kung University, Tainan 70428, Taiwan; 5Department of Medical Research, China Medical University Hospital, China Medical University, Taichung 40402, Taiwan

**Keywords:** silver nanoparticles, zinc oxide nanoparticles, developmental toxicity, reactive oxidative stress, apoptosis, autophagy

## Abstract

The global application of engineered nanomaterials and nanoparticles (ENPs) in commercial products, industry, and medical fields has raised some concerns about their safety. These nanoparticles may gain access into rivers and marine environments through industrial or household wastewater discharge and thereby affect the ecosystem. In this study, we investigated the effects of silver nanoparticles (AgNPs) and zinc oxide nanoparticles (ZnONPs) on zebrafish embryos in aquatic environments. We aimed to characterize the AgNP and ZnONP aggregates in natural waters, such as lakes, reservoirs, and rivers, and to determine whether they are toxic to developing zebrafish embryos. Different toxic effects and mechanisms were investigated by measuring the survival rate, hatching rate, body length, reactive oxidative stress (ROS) level, apoptosis, and autophagy. Spiking AgNPs or ZnONPs into natural water samples led to significant acute toxicity to zebrafish embryos, whereas the level of acute toxicity was relatively low when compared to Milli-Q (MQ) water, indicating the interaction and transformation of AgNPs or ZnONPs with complex components in a water environment that led to reduced toxicity. ZnONPs, but not AgNPs, triggered a significant delay of embryo hatching. Zebrafish embryos exposed to filtered natural water spiked with AgNPs or ZnONPs exhibited increased ROS levels, apoptosis, and lysosomal activity, an indicator of autophagy. Since autophagy is considered as an early indicator of ENP interactions with cells and has been recognized as an important mechanism of ENP-induced toxicity, developing a transgenic zebrafish system to detect ENP-induced autophagy may be an ideal strategy for predicting possible ecotoxicity that can be applied in the future for the risk assessment of ENPs.

## 1. Introduction

Currently, engineered nanomaterials and nanoparticles (ENPs) are used in a wide variety of industrial and commercial applications, including catalysts, environmental remediation, personal care products, and cosmetics. In addition, ENPs also show great promise in medicine, such as imaging and drug delivery [1]. Examples of two widely utilized ENPs are silver nanoparticles (AgNPs) and zinc oxide nanoparticles (ZnONPs) [2]. AgNPs are used in cosmetics, textiles, antibacterial agents, the food industry, paints, and medical devices, whereas ZnONPs are widely used in sunscreen, cosmetics, paints, and antibacterial ointments [3,4]. The global usage of both AgNPs and ZnONPs has increased exponentially, thereby increasing the amount of AgNPs and ZnONPs that enter the aquatic environment through various routes [3]. The continuous release of ENPs into the environment from consumer products, industrial waste, and sewage sludge raises concerns about their distribution, behavior, and characteristics in natural water and adverse effects on ecosystems [5]. Nevertheless, the comprehensive toxicological effects of AgNPs and ZnONPs present in the environment have still not been studied in detail. Thus, an analysis of the effects of AgNPs and ZnONPs on ecological systems and human health is important.

In general, a detailed understanding of the behavior of ENPs in aquatic environments is crucial to determine their final distribution and the associated risks [5,6,7]. Due to their small size, some ENPs, such as AgNPs and ZnONPs, release ions in aquatic systems that are responsible for the major causes of toxicity. However, they also have a confined solid phase similar to other poorly soluble compounds in the aquatic system [7]. In a real water system, natural organic matter (NOM), certain biological macromolecules, and the environmental conditions, including pH and ionic strength, will modify the ENP behavior [5]. Predicting the fate of ENPs in natural surface water systems continues to be a challenge because ENPs can heteroaggregate with natural organic and mineral suspended matter. For example, NOM has been proposed to dominate ENP surface chemistry, and inorganic compounds such as sulfate also play important roles in the modifications of ENP structures and in reducing their toxicity [5]. Transformations of ENPs, such as agglomeration, aggregation, dissolution, and surface change, by interacting with natural water components might substantially alter their environmental release and toxicity [2]. Thus, defining the behavior of ENPs in aquatic systems and their toxic effects on living organisms can facilitate the establishment of scientific foundations for the risk assessment of ENPs in aquatic ecosystems.

The zebrafish (*Danio rerio*) embryo (ZFE) is an ideal model to assess the hazards of both conventional chemicals and ENPs in (eco)toxicology [8]. This model possesses advantages, including rapid development and optical transparency, allowing easy observations of phenotypic responses at lethal, acute, and sublethal toxicological endpoints. In addition, large amounts of embryos can be generated rapidly at low cost, permitting them to serve as a high-throughput assay for the study of developmental processes upon exposure to ENPs [8]. We have recently reported the possible targets and mechanisms of the toxic effects of AgNPs on a ZFE model. Our major finding is that exposure to AgNPs alters lysosomal activity (an indicator of autophagy) and leads to a greater number of apoptotic cells distributed among the developmental organs of the embryo [8,9]. From the (eco)toxicological perspective, the toxic effects induced by AgNPs and ZnONPs, including cytotoxicity, hematotoxicity, immunotoxicity, hepatotoxicity, and embryotoxicity, either in vitro or in vivo, are quite similar [10]. The well-known toxicity induced by ZnONPs is predominantly mediated by the formation of ROS. Excessive ROS generation may damage mitochondria, which subsequently leads to inflammasome activation and cell death through apoptosis and autophagy, which might be a novel mechanism modulating ZnONP-induced inflammatory and cytotoxic effects [11]. ENPs-induced oxidative stress is considered the initiator of the disruption of mitochondrial membrane potential and apoptosis and/or autophagy. Although extensive research has been conducted on ENP applications and toxic mechanisms, research on the environmental transport behavior and ecotoxicity of emerging materials such as AgNPs or ZnONPs is still limited and is needed for sustainable environmental implementation.

We conducted experiments in both complex natural original waters (lake, reservoir, and river) and filtrates from natural waters produced with different pore sizes of filters (1 µm, 0.45 µm, 0.22 µm, and 0.1 µm) to understand the behavior, characteristics, and embryotoxicity of AgNPs and ZnONPs in aquatic systems. Although results obtained from natural waters may more realistically approximate actual ecosystems, they are often unable to provide sufficient information due to the complexity of water chemistry. The results obtained from filtrates of natural waters and MQ water (control group), which have a reduced complexity, can improve our understanding of the processes involved but are not necessarily representative of natural systems. The objective of this study was to characterize aggregates of AgNPs and ZnONPs in natural waters and determine whether they are toxic to developing zebrafish embryos. Different toxic effects and mechanisms were investigated here to evaluate ENP toxicity by measuring the survival rate, hatching rate, body length, oxidative stress, apoptosis, and autophagy. The results from those experiments are essential for a better understanding of AgNPs and ZnONPs in aquatic ecosystems and provide important underlying mechanisms for ecological risk assessments of them and other nanoparticles.

## 2. Materials and Methods

### 2.1. Natural Water Preparation and Sampling

We collected natural water samples from different water bodies, including the Erren River, Zengwun Reservoir, and Cheng Kung Lake located at the campus of National Cheng Kung University (NCKU), in spring (May) and autumn (September) in Taiwan. Among them, the Erren River was considered a polluted water body, whereas Zengwun reservoir water was the source of drinking water for the public. Natural water was collected in sterilized sampling bags. Before storage, all the original water samples were filtered with a 53 μm filter to remove large impurities and microorganisms, and then the water samples were stored at 4 °C.

### 2.2. Preparation and Characterization of AgNPs and ZnONPs

The principle of AgNP synthesis was based on the NaBH_4_ reduction of the Ag ion. First, the silver nitrate solution (50 mL, 20 mM), sodium citrate solution (40 mL, 80 mM). and 890 mL deionized water were fully mixed. Then, a NaBH_4_ solution (20 mL, 100 mM) was added slowly to promote the Ag ion reduction. The solution was stirred vigorously for 2 h, allowing a full reductive reaction. After synthesizing a sufficient amount of AgNP solution, it was centrifuged at 7500× *g* for 30 min, and the supernatant was collected. Furthermore, the solution was centrifuged again at 12,500× *g* for 2 h, and the precipitate was removed. The synthesized AgNP solution was stored at 4 °C in the dark.

We dissolved zinc acetate dehydrate (Zn(CH_3_COO)_2_(H_2_O)) in 3.35 mM ethanol and stirred the solution vigorously at 60 °C to synthesize ZnONPs. Furthermore, the solution was slowly titrated with a 6.59 mM potassium hydroxide (KOH) solution into a zinc acetate dehydrate solution along the edge of the beaker and reacted for 1.5 h. Next, the solution was incubated at room temperature for 2 h. The synthetic ZnONP solution was centrifuged at 10,000 rpm for 10 min, and then the supernatant was removed. The precipitate was collected and subsequently washed twice with 50 mL of ethanol. Then, 0.25 mL of (3-aminopropyl)triethoxysilane (APTES), 0.5 mL of deionized water, and 0.05 mL of ammonia (25 wt%) were mixed immediately with the ZnONP solution and reacted at room temperature for 20 min. After the reaction, the solution was centrifuged at 10,000 rpm for 15 min, and then we collected the precipitates. The precipitates were washed 2 times with ethanol and resuspended in water to obtain amine-coated ZnONPs. The aim of the surface modification was to improve the dispersion of nanoparticles in various solutions. Precisely, the formal name of ZnONPs we applied in this study was aminopropyl silica-coated ZnONPs, which is abbreviated as ZnONPs. The primary size and morphology of NPs were observed by transmission electron microscopy (TEM, JEOL Co., Akishima, Tokyo, Japan). The elemental analysis of NPs and NPs spiked in natural water were detected through electron dispersive X-ray (EDX) (JEOL Co., Akishima, Tokyo, Japan). The hydrodynamic diameter and polydispersity index were analyzed via dynamic light scattering (DLS, Delsa™ Nano C, Beckman Coulter, Inc., Brea, CA, USA). The zeta potential of the NPs was measured by phase analysis light scattering (PALS, Delsa™ Nano C, Beckman Coulter, Inc., Brea, CA, USA). The NP stability was investigated by ultraviolet visible spectroscopy (UV-Vis, Thermo Fisher Scientific, Waltham, MA, USA).

### 2.3. Fish Husbandry and Egg Spawning

Zebrafish embryos (*Danio rerio*) were obtained from the Taiwan Zebrafish Core Facility. Zebrafish were raised and maintained in a thermostatic culture system at 28–30 °C with a photoperiod of 14 h of light/10 h of darkness. Fish were fed twice a day with brine shrimp. Male and female fish were placed in the mating box. Spawning was triggered when the light was turned on in the morning. At 4 hpf, zebrafish embryos were collected and rinsed several times to remove residues on the surface. The dead, unfertilized, and abnormal embryos were removed. Healthy embryos were randomly selected for experiments.

### 2.4. The Acute Toxicity Test on Zebrafish Embryos

The fish embryo acute toxicity test conducted in this study was based on OECD Test Guideline TG No. 236, released in 2013. All experiments were performed using a semi-static system. First, the fertilized embryos were gently collected with a dropper at 4 hpf, and healthy embryos were randomly divided and placed in 12-well plates (10 embryos/well). Embryos (30 embryos/group) were treated with AgNPs and ZnONPs. The positive controls (4 mg/L 3,4-dichloroaniline) were observed until 96 hpf, and mortality and deformities were recorded daily. The AgNPs and ZnONPs solutions were refreshed every 24 h until the end of the experiment to avoid ENP aggregation during exposure. All acute toxicity assays were performed in triplicate. 

### 2.5. ROS Analyses

Dichlorodihydrofluorescein diacetate (DCFH-DA) (Sigma-Aldrich, St. Louis, MO, USA) was used to measure reactive oxygen species (ROS) levels in the zebrafish. The zebrafish embryos were exposed to AgNPs and ZnONPs solutions beginning at 4 hpf, and the DCFH-DA assay was conducted at 96 hpf. In the experimental procedure, we initially performed three washes with deionized water to remove solutions of AgNPs and ZnONPs and impurities from the surface of the zebrafish. Then, live samples were incubated with 5 μg/mL DCFH-DA at 28 °C for 1 h and coated with aluminum foil to protect them from light. After the incubation, the living fish were rinsed with deionized water three times. The embryos were fixed with 3% methyl cellulose to prevent movement, and images were captured immediately.

### 2.6. LysoTracker RED DND-99 Analyses 

LysoTracker RED DND-99 (Sigma-Aldrich, St. Louis, MO, USA) reagent was used to evaluate lysosomes and autophagy in the zebrafish. The procedure was performed according to the manufacturer’s protocol. Briefly, the samples were collected at 72 hpf and rinsed with deionized water three times. Then, the embryos were incubated with 10 μM lysosomal probe for 1 h at 28 °C. After the incubation, we used deionized water to wash the embryos three times and then fixed the embryos with 3% methyl cellulose. Embryos were observed and imaged using a fluorescence microscope (BX51, Olympus Co., Tokyo, Japan).

### 2.7. Whole-Mount TUNEL Assay

Whole-mount TUNEL staining was performed to evaluate apoptosis in zebrafish at 72 hpf. We washed the zebrafish with deionized water three times after exposure to Ag/ZnONPs. Zebrafish larvae were fixed with 4% paraformaldehyde for 1 h and then washed with PBS three times. Next, the embryos were incubated with blocking buffer on a shaker for 30 min and subsequently rinsed with PBS three times for 5 min. Then, the embryos were incubated with a permeabilization solution (0.1% Triton X-100 in 0.1% sodium citrate) for 30 min on ice and washed with PBS three times on a shaker for 5 min each. Finally, the TUNEL reaction mixture (labeling solution:enzyme solution = 9:1) was incubated with embryos in a water bath at 37 °C for 1 h, and we observed the embryos using a confocal fluorescence microscope (Nikon TE2000EPFS-C1-Si, Tokyo, Japan).

### 2.8. Statistics

Each experiment was independently repeated at least three times. Statistical analyses were performed with one-way ANOVA and a two-tailed Student’s *t*-test. *p* < 0.05 was considered statistically significant between the experimental and control groups. All data were presented as the mean ± SEM.

## 3. Results

### 3.1. The Physical and Chemical Properties of AgNPs and ZnONPs

Two types of nanoparticles were synthesized in this study to evaluate the environmental toxicity of ENP-spiked natural water. We synthesized citrate-coated AgNPs and amine-modified ZnONPs and fully characterized their physicochemical properties. As shown in Figure 1, The TEM images present the mean primary sizes of AgNPs and ZnONPs, which were 13 and 27 nm, respectively, and the morphology was spherical (Figure 1a,c). Element analysis indicated that AgNPs contained silver and ZnONPs were composed of zinc and oxygen (Figure 1b). The Si element came from a synthesized process which used APTES, and the Cu came from carbon support films on copper grids, which were used in TEM analysis. Dynamic light scattering (DLS) results revealed that the hydrodynamic diameters of the AgNPs and ZnONPs were 16 nm and 43 nm, respectively, and the NPs exhibited a homogeneous dispersion (Figure 1c). In addition, AgNPs and ZnONPs represented negative (−18.6) and positive surface charges (+25.4), respectively (Figure 1c).

### 3.2. The Deposition of AgNPs and ZnONPs Spiked in Natural Water

Several studies have shown that the physicochemical characteristics of ENPs change when they are spiked in natural water bodies [12,13]. We obtained water from NCKU Lake, Zengwun Reservoir, and Erren River to determine the physicochemical properties of Ag/ZnONPs in natural water. NCKU Lake is located at National Cheng Kung University and possesses a high species richness. Zengwun Reservoir is a relatively clean water source that provides drinking water. The Erren River is one of the contaminated rivers in Taiwan. To date, the Erren River is still considered moderately polluted. 

AgNPs and ZnONPs were spiked into the water samples from NCKU Lake, Zengwun Reservoir, and Erren River, and deposition was observed after 30 min. For the NCKU Lake water sample, the AgNPs were still suspended in water, but the color quickly changed. ZnONPs spiked in NCKU Lake water were deposited. In addition, water from the Zengwun Reservoir and Erren River caused a substantial aggregation of AgNPs and ZnONPs upon spiking into the natural water (Figure 1a,b). Furthermore, the physicochemical characteristics of these nanoparticles spiked in natural water were analyzed by determining their particle size, hydrodynamic diameter, and dispersity, as summarized in Figure 2c–e. Both AgNPs and ZnONPs spiked in NCKU Lake water exhibited a larger particle size and hydrodynamic diameter and decreased dispersity (Figure 2c,e). After spiking in the Zengwun Reservoir and Erren River water, AgNPs and ZnONPs aggregated and were not suspended in solution, and thus the data were undetectable (Figure 2c,e). These results were attributed to the chemical reaction of these nanoparticles with complex water components after spiking in natural water, which led to aggregation and precipitation.

### 3.3. Elemental Mapping of Natural Water 

Furthermore, we performed elemental mapping to determine the elements present in natural water that reacted with AgNPs and ZnONPs. NCKU Lake, Zengwun Reservoir, and Erren River water samples were spiked with AgNPs and ZnONPs. We found that NCKU Lake, Zengwun Reservoir, and Erren River water precipitants contained abundant sulfur (S) and chlorine (Cl) and that Ag perfectly colocalized with them (Figure 3a–c). For ZnONP-spiked water samples, zinc was accompanied by magnesium (Mg) and potassium (K) in NCKU Lake water (Figure 3d). Furthermore, Zengwun Reservoir water samples contained calcium (Ca) and chlorine that colocalized with zinc (Figure 3e). Last, the Erren River water contained sulfur and calcium that overlapped with zinc (Figure 3f). Based on these results, we determined that both AgNPs and ZnONPs undergo complex chemical reactions with the components in natural water.

### 3.4. The Mortality of AgNPs and ZnONPs Spiked in Nature Water

First, we observed the survival of embryos after exposure to the three water samples to determine whether natural water exerted adverse effects on embryos. Compared with the positive control group (3,4-DCA), the original water samples that were not spiked with AgNPs and ZnONPs did not cause obvious mortality at 24, 48, and 96 hpf. Even in the sample from the polluted Erren River, the survival rate was still high (Appendix A). When the original water samples were filtered through membranes with different pore sizes of 1 µm, 0.45 µm, 0.22 µm, and 0.1 µm, the filtrates also did not produce significant adverse effects. The survival rates of all groups were still greater than 80% (Appendix A).

According to previous studies, natural substances, such as humic acid, in natural water mitigated the toxic effects of NPs [13]. We designed additional experiments to confirm whether the substances in natural water resulted in a reduction in toxicity. First, the larger substances in natural water were removed with a 0.45 µm filter in advance and then the samples were spiked with AgNPs and ZnONPs (Figure 4a,b). Second, the original natural water was spiked with AgNPs and ZnONPs first and then passed through a 0.45 µm filter (to remove deposited ENPs) (Figure 4a,b). Third, we spiked 1 and 10 µg/mL AgNPs and ZnONPs in MQ water, respectively, for comparison and observed a substantial increase in mortality (Figure 4c,d). Accordingly, ZnONPs spiked in the natural water samples from NCKU Lake, Zengwun Reservoir, and Erren River exerted minor toxic effects, even in the 10 µg/mL groups (Figure 4e,f). In terms of the deposited NP removal groups, the elimination of AgNPs and ZnONPs deposits in natural water reversed the mortality, especially in the 10 µg treatment group (Figure 4g–j). In summary, although AgNPs and ZnONPs spiked into natural water exhibited a dramatically reduced toxicity, they still exerted adverse effects on the ecological environment at higher concentrations.

### 3.5. ZnONPs Caused Developmental Toxicity

We measured body length to explore the relationship between zebrafish embryo development and substances in the natural water environment. Interestingly, the filtrates of water samples from NCKU Lake, Zengwun Reservoir, and Erren River that were passed through 1 μm, 0.45 μm, 0.22 μm, and 0.1 μm filters did not alter the body length of the zebrafish compared with the control group (MQ water) (Appendix A). The body length of the zebrafish even increased after exposure to filtrates of natural water compared to the control group, but significant differences were not observed between the filtrate groups. In addition to the survival rate and body length, the hatching rate is also an indicator of developmental toxicity. When ZnONPs were spiked into the three different natural water samples and then passed through a 0.45 μm filter, we observed a significantly later hatching time at 96 hpf in embryos exposed to a higher concentration of 10 μg/mL (Figure 5b). Nonetheless, the embryos exposed to ZnONPs still hatched successfully without deformities. The AgNPs did not induce a delayed hatching time in this study (data not shown).

### 3.6. AgNPs and ZnONPs Induced Programmed Cell Death 

Several studies have revealed that AgNPs and ZnONPs lead to cytotoxicity by inducing programmed cell death [8,14,15]. We applied the TUNEL assay to investigate the cytotoxic effects of AgNPs and ZnONPs spiked in deionized (MQ) or natural water at 72 hpf. DNase I served as a positive control group and resulted in a significant increase in the number of apoptotic cells (Figure 5a,b). Groups treated with AgNPs and ZnONPs spiked in the MQ water exhibited an increased number of apoptotic cells, and the signals were mainly located in the yolk sac (Figure 5a,b). Surprisingly, groups treated with both AgNPs and ZnONPs spiked in natural water also exhibited substantial cytotoxicity. These results provide evidence that ENPs in natural water exert cytotoxic effects by activating apoptosis.

### 3.7. AgNPs and ZnONPs Induced ROS Production

Excessive oxidative stress is an important cellular mechanism of ENPs and is recognized as an initial form of cell damage [8,14,16,17]. Next, we investigated whether AgNPs and ZnONPs spiked in different water samples would increase the ROS level in zebrafish embryos. We used dichloro-dihydro-fluorescein diacetate (DCFH-DA) dye to detect ROS generation in embryos exposed to these nanoparticles at 72 hpf. As expected, the H_2_O_2_ groups (positive control) displayed increased ROS generation (Figure 6a,b). Similarly, 10 µg/mL AgNPs and ZnONPs spiked in the MQ water increased ROS generation, and the signals were primarily observed in the intestine (Figure 6a,b). As mentioned above, we concluded that ENPs spiked into the natural water increase ROS level in zebrafish embryos. 

### 3.8. AgNPs and ZnONPs Induced Autophagy 

According to recent studies, AgNPs and ZnONPs cause dysfunctional autophagy in vitro [14,18]. Hence, we used the LysoTracker RED probe to evaluate lysosomal activity in zebrafish embryos incubated with AgNPs and ZnONPs spiked in different water samples. Treatment with 10 μg/mL AgNPs and ZnONPs in the MQ water activated autophagy, and the signals were detected in the yolk sac (Figure 7a,b). For natural water, AgNPs spiked in Zengwun Reservoir natural water apparently increased the activation of autophagy (Figure 7a). Some lysosomal activity was also observed in the other groups. Therefore, AgNPs and ZnONPs spiked in natural water induce autophagy.

## 4. Discussion

Engineered nanomaterials and nanoparticles (ENPs) emission are estimated to be mainly derived from landfills (63–91%), with over 260,000–309,000 metric tons of global production in 2010, followed by release to soil (8–28%), water bodies (0.4–7%), and air (0.1–0.5%) [19]. In certain environmental compartments, ENPs may pose a relatively low risk, whereas organisms residing near ENP production plant outfalls or waste treatment plants may be at increased risk [20]. When ENPs enter terrestrial and aquatic systems, they may threaten the ecological environment and human health [21]. In general, the fate of ENPs in aquatic systems is mainly determined by three processes: heteroaggregation, dissolution, and sedimentation [22,23]. The factors that influence ENP behavior in the environment include size, surface coating materials and their changes (e.g., degradation or replacement by natural organic matter (NOM)), oxidation, dissolution, sulfidation, heteroaggregation, homoaggregation, and sedimentation/resuspension [24,25,26]. Other environmental factors include sunlight, the pH of the solution, inorganic salts, the interaction with surrounding metals, and dissolved NOM, which will interact with ENPs and lead to their transformation [27,28,29,30,31]. 

After AgNPs and ZnONPs were added to natural water, we observed that the color and particle size of the nanoparticles changed significantly (Figure 1 and Figure 2). Thus, the nanoparticles may undergo chemical reactions with other components in the water environment. The dissolution of nanoparticles might be generally promoted after interacting with NOM. Two major surface transformation processes, oxidation and sulfidation, may occur on the surface of nanoparticles in the presence of NOM [32,33]. Therefore, NOM alters the toxicity of ENPs by changing suspension stabilization, the bioavailability of metal ions, electrostatic interactions and steric repulsion between nanoparticles and organisms, and induced reactive oxygen generation [33]. Bundschuh et al. noted that the phenomenon of co-occurring contaminants interacting with nanoparticles and indicated that nanoparticles serve as a sink for organic and inorganic co-contaminants in the water column [34]. Therefore, we conducted an elemental mapping analysis to investigate the composition of sediments of AgNPs or ZnONPs after their addition to natural water. Sulfur (S) and chlorine (CI) attached to AgNPs and ZnONPs, and iron (Fe) and phosphorous (P) attached on ZnONPs, which may in turn change the physicochemical properties of the nanoparticles due to the interaction between these molecules (Figure 3). AgNPs release silver only after they are oxidized by dissolved oxygen, and the released silver is readsorbed onto the surface of the nanoparticles or forms a secondary precipitate with complexing species (e.g., Cl^−^ and SO_4_^2−^) [35]. Sulfidation of AgNPs or ZnONPs frequently occurs under various environmental conditions and leads to the formation of core–shell Ag0–Ag2S structures or hollow Ag2S NPs. Sulfidation leads to nearly inert NP surfaces that alter their reactivity and toxicity [34]. The fate and stability of nanoparticles in both raw lake water and filtered lake water containing different NOM lead to different aggregation profiles [36]. The authors concluded that the use of pure NOM analogs may not accurately represent nanoparticles’ interactions and fate in real natural systems [36]. Our experiment results suggested that the natural water may have mitigated the toxic effects of AgNPs and ZnONPs through nanoparticles aggregation and interaction with NOM, resulting in the formation of larger particles and sedimentation. Nonetheless, the underlying mechanisms of the interactions and relationships among nanoparticles and organic/inorganic substances in the ecosystem require further investigation.

With the advantages of rapid development and optical transparency, the zebrafish embryos are rapidly becoming an attractive vertebrate model species for screening ENPs [37]. Our current study showed a very high survival rate of zebrafish embryos exposed to three different original natural waters samples and their filtrates obtained after passing through different pore sizes of filter. Interestingly, all the above-mentioned natural water samples led to a longer body length of larva than the embryos exposed to MQ water (Appendix A). On the one hand, natural water samples may contain certain essential elements that enhance the development of the embryos. On the other hand, chemicals and/or ENP contamination of surface waters from rivers, lakes, and reservoirs in Taiwan may still be limited and promote the survival of zebrafish embryos. Currently, aquatic AgNP concentrations in fresh water are predicted to range from approximately a few pg/L to 10 ng/L between 2017 and 2050, which might be nontoxic to fish embryos [20]. Although the current ENP contamination may pose a relatively low risk to natural aquatic systems, the organisms living in the ecosystem near ENP production plants or waste treatment plants may be at higher risk. Therefore, we conducted acute zebrafish embryo toxicity assays by spiking AgNPs or ZnONPs into natural water samples. As shown above (Figure 6, Figure 7 and Figure 8), both AgNPs and ZnONPs led to significant acute toxicity toward zebrafish embryos in a dose-dependent manner. The level of acute toxicity was relatively lower in the filtered natural water samples than in the MQ water samples, indicating that the interaction and transformation of these nanoparticles with the complex components in a water environment led to a reduced toxicity.

Zebrafish embryotoxicity tests have been indicated as a suitable approach for assessing the toxicity of both traditional chemicals and ENPs [38,39]. Nonetheless, the majority of the published studies were conducted in the laboratory with controlled standard water samples. Here, we aimed to reveal the potential toxic effects and mechanisms of AgNPs and ZnONPs on zebrafish embryos in natural water. One of the interesting findings is that ZnONPs, but not AgNPs, triggered a significant delay in embryo hatching (Figure 5). Consistent with our finding, Chen. et al. reported that exposure to ZnONPs suspensions and their respective centrifuged supernatants caused similar hatching delays, whereas the supernatants did not cause larval mortality or malformation. In addition, coexposure to N-acetylcysteine (NAC), a well-known antioxidant, did not alter the effects of ZnONPs on hatchability but rescued their behavioral effect [40]. Thus, the toxicity of ZnONPs may be due to a combination of the effects of dissolved Zn ions and particle-induced oxidative stress. Zinc is an essential transition metal in living organisms that plays an important role in the maintenance of protein structure and enzymatic function. However, excessive free Zn ions are toxic and may be bound by Zn-binding proteins, such as metalloproteins [41]. The dissolved Zn ions interfere with embryo hatching through a chelator-sensitive mechanism that involves the ligation of histidines in the metalloprotease ZHE1, which is responsible for degradation of the chorionic membrane [42,43]. The effects of ZnONPs on delaying hatching were attenuated in filtered natural water samples compared with MQ water, suggesting the slow release of Zn ions from ZnONPs and the interaction of dissolved Zn ions with the complex components in the water environment, which subsequently mitigate effects on embryonic development. Surface coating with different chemicals or NOM may influence the colloidal stability and solubility of ZnONPs or AgNPs and thereby modulate toxicity [44]. 

Both ENP-induced mortality and developmental toxicity seem to be related to oxidative stress. Excess ROS production may contribute to tissue damage and participate in signal transduction, the proliferative response, gene expression, and protein redox regulation [45]. ENP-induced oxidative stress was proposed as one of the initiators of the disruption of the mitochondrial membrane potential, the induction of ER stress, and cell death mediated by apoptosis and/or autophagy [9]. The mechanisms underlying ENP-induced toxicity have become one of the most frequently studied topics in toxicology during the last two decades. Our previous studies were the first to show that autophagy activated by AgNPs fails to trigger the lysosomal degradation pathway and leads to dysfunctional autophagy, which is relevant to the accelerated cellular pathogenesis of diseases [10,45,46]. More recently, we also prioritized the factors affecting the toxic potential of AgNPs, which included exposure dose/time, cell type, and the size and surface coating of AgNPs. Using an in silico decision tree-based knowledge discovery-in-databases process, the toxicity-related parameters are ranked as follows: exposure dose > cell type > particle size > exposure time ≥ surface coating [47]. AgNPs with larger particle sizes appeared to induce higher levels of autophagy during the earlier phase of both subcytotoxic and cytotoxic exposures in the in vitro cell culture models, whereas apoptosis, but not necrosis, accounted for the compromised cell survival over the same dosage range [47]. In addition, we determined the skin toxicity and the potential mechanisms of ZnONPs combined with UVB exposure and the preventive effect of a well-known antioxidant, pterostilbene. Exposure to both ZnONPs and UVB disrupts cellular autophagy, which in turn increases exosome release from cells. Application of the antioxidant pterostilbene reversed autophagy abnormalities by restoring normal autophagy flux and decreasing NLRP3 inflammasome-loaded exosome release through the attenuation of total ROS and mitochondrial ROS levels [11]. In general, autophagy is a cellular recycling pathway by which lysosomes degrade damaged organelles and/or proteins to maintain cellular homeostasis. However, ENPs have been proven to induce autophagic cell death in several cell types by interfering with autophagy flux and disrupting lysosomal function [48]. The leakage of lysosomal enzymes activates procaspases or damages the mitochondrial outer membrane to induce apoptosis. As shown in the present study, zebrafish embryos exposed to filtered natural water spiked with AgNPs or ZnONPs presented increased ROS levels, apoptosis, and lysosomal activity, an indicator of autophagy (Figure 7 and Figure 8). To the best of our knowledge, the induction of autophagy in zebrafish embryos triggered by ENPs in natural water has seldom or never been reported previously. As human being and ecosystem exposure to ENPs is unavoidable, an in-depth understanding of ENP-modulated autophagy is required to assess their safety [48]. 

The existing literature on the embryotoxicity and teratogenicity of ENPs in zebrafish has been reported in a recent review article [39]. The interaction and bioaccumulation of ENPs in zebrafish embryos are associated with several toxic effects, such as delayed hatching, yolk sac alterations, circulatory changes, and musculoskeletal disorders. In addition, the toxic effects of ENPs on innate immunity in a zebrafish model have also been reported [49]. Most of the abovementioned toxic effects are related to dysregulated autophagy. Since autophagy is considered an early indicator of ENP interactions with cells and has been recognized as an important form of cell death in ENP-induced toxicity, creating an autophagy-related transgenic zebrafish line could be a good approach to monitor the ENP pollution in an ecosystem. Overall, our study revealed that AgNPs and ZnONPs spiked in natural water increased zebrafish embryo mortality at higher concentrations, delayed the hatching rate, and induced ROS production, autophagy, and apoptosis (Figure 9). Our current study focused on AgNPs and ZnONPs, which are widely used in several industries, and described their behavior, characteristics, embryotoxicity, and underlying mechanisms in natural aquatic systems. These results will enable the development of more relevant testing methods to predict the possible long-term ecotoxicity of ENPs and can be applied in the future for regulatory decision-making and risk assessments of ENPs.

## 5. Conclusions

Our research confirmed that nanoparticles of AgNPs and ZnONPs spiked in NCKU Lake, Zengwun Reservoir, and Erren River cause minor toxic effects. We speculated that AgNPs and ZnONPs spiked in the natural water had a stronger aggregation and changed their physicochemical properties by interacting with the surrounding environment, which consequently mitigated their toxic effects. However, AgNPs and ZnONPs spiked in the natural water increased the mortality of zebrafish embryos and delayed hatching time in higher concentrations, as well as induced ROS, autophagy, and apoptosis.

## Figures and Tables

**Figure 1 nanomaterials-12-00717-f001:**
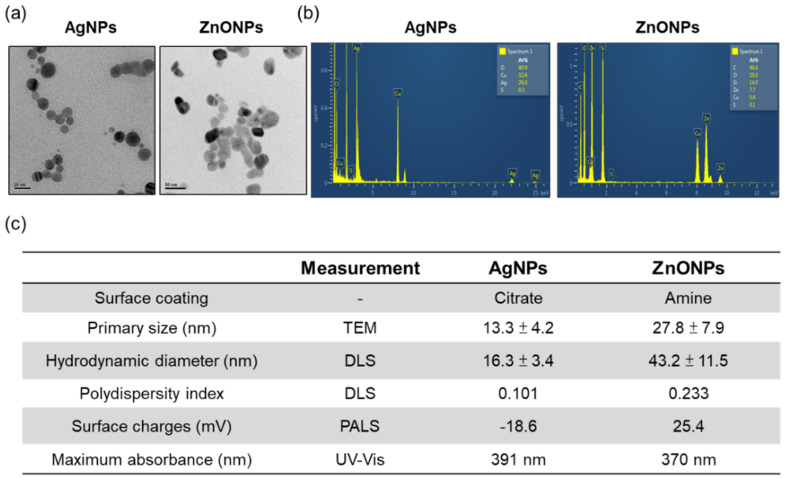
Physical and chemical properties of AgNPs and ZnONPs. (**a**) TEM images of AgNPs and ZnONPs. (**b**) EDX analysis of AgNPs and ZnONPs. (**c**) Analysis of the physicochemical properties of AgNPs and ZnONPs. AgNPs, silver nanoparticles; ZnONPs, zinc oxide nanoparticles; TEM, transmission electron microscopy; EDX, electron dispersive X-ray; DLS, dynamic light scattering; PALS, phase analysis light scattering; UV–Vis, ultraviolet visible spectroscopy.

**Figure 2 nanomaterials-12-00717-f002:**
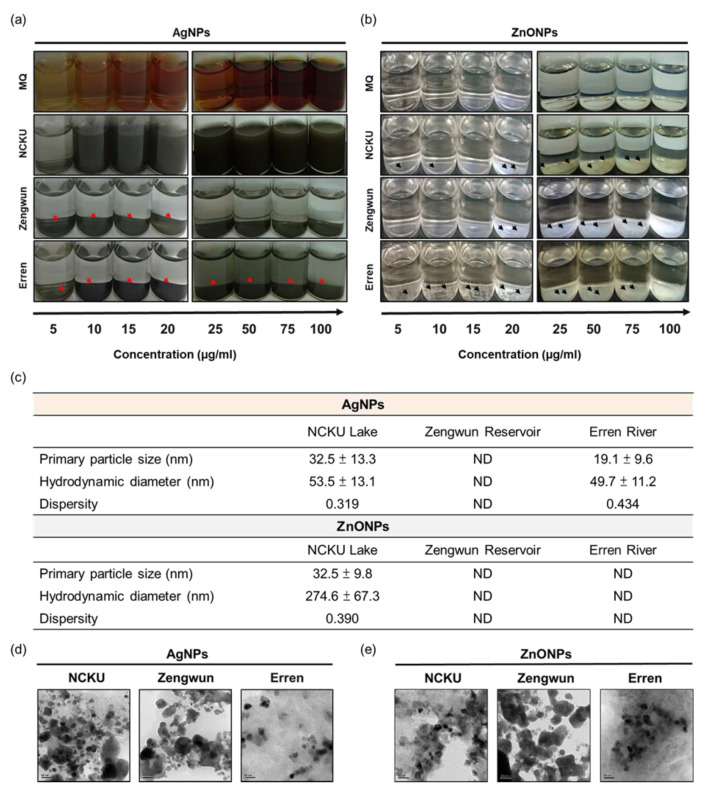
Deposition of AgNPs and ZnONPs spiked in natural water. The deposition of 5, 10, 15, 20, 25, 50, 75, and 100 µg/mL (**a**) AgNPs and (**b**) ZnONPs spiked in natural water from NCKU Lake, Zengwun Reservoir, and Erren River. Ag/ZnONPs aggregated and deposited at the bottom of the vessel. The arrow indicates Ag/ZnONPs deposition. (**c**) The physical and chemical properties of AgNPs and ZnONPs spiked in natural water. (**d**,**e**) TEM images of AgNPs and ZnONPs spiked in natural water. AgNPs and ZnONPs aggregated after spiking in natural water samples. AgNPs, silver nanoparticles; ZnONPs, zinc oxide nanoparticles; TEM, transmission electron microscopy.

**Figure 3 nanomaterials-12-00717-f003:**
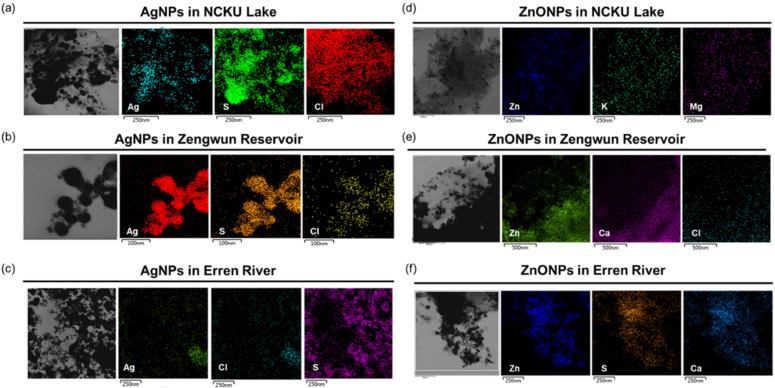
Elemental mapping images of AgNPs and ZnONPs in natural water. (**a**–**c**) Elemental mapping analysis of Ag spiked in NCKU Lake, Erren River, and Zengwun Reservoir water. (**d**–**f**) Elemental detection of ZnONPs spiked in natural water. AgNPs, silver nanoparticles; ZnONPs, zinc oxide nanoparticles; S, sulfur; Cl, chlorine; Mg, magnesium; K, potassium; Ca, calcium.

**Figure 4 nanomaterials-12-00717-f004:**
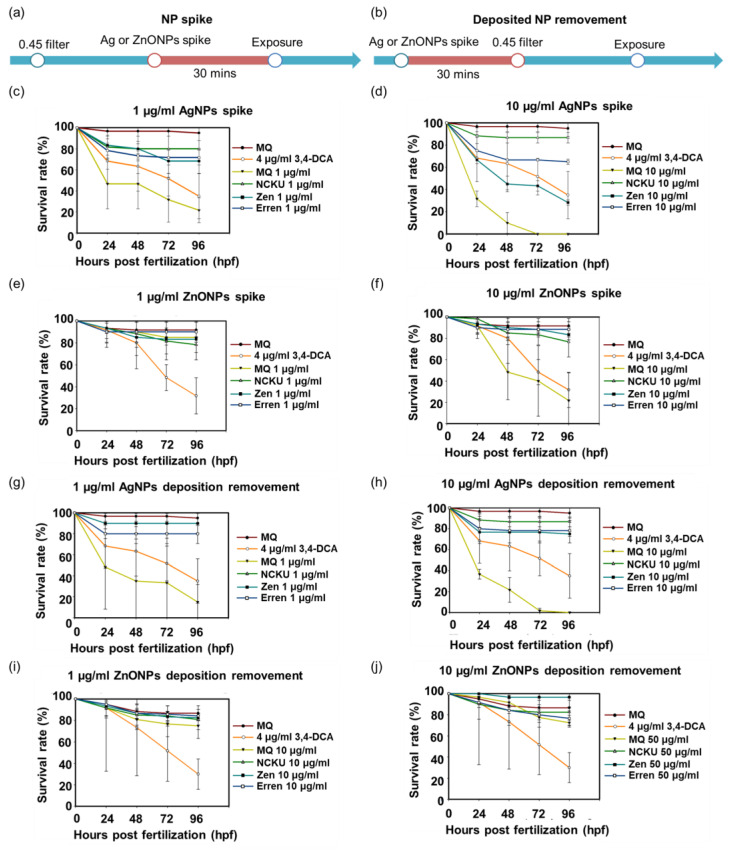
Lethality of AgNPs and ZnONPs spiked in natural water. The study design included two processes: (**a**) NP spiking and (**b**) deposited NP removal. The embryos were treated with (**c**) 1 µg/mL AgNPs, (**d**) 10 µg/mL AgNPs, (**e**) 1 µg/mL ZnONPs, and (**f**) 10 µg/mL ZnONP-spiked natural water. The mortality of embryos exposed to (**g**,**h**) 1 µg/mL and 10 µg/mL AgNP-spiked water, as well aI (**i**,**j**) 1 µg/mL and 10 µg/mL ZnONP-spiked natural water in which the aggregates were removed by a 0.45 μm filter. AgNPs, silver nanoparticles; ZnONPs, zinc oxide nanoparticles; 3,4-DCA, 3,4-dichloroaniline.

**Figure 5 nanomaterials-12-00717-f005:**
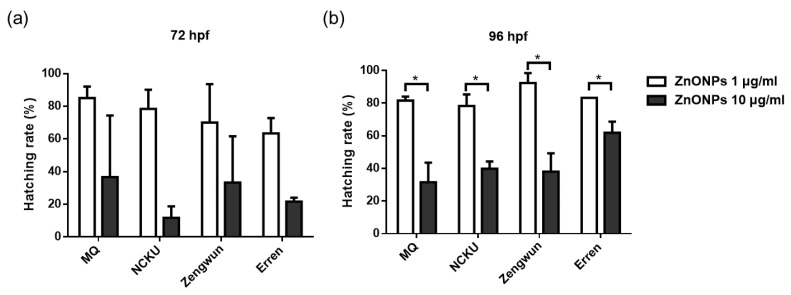
Developmental toxicity of ZnONPs. Hatching rate of zebrafish embryos exposed to ZnONPs. ZnONPs spiked in NCKU Lake, Zengwun Reservoir, and Erren River water. Hatching rate at (**a**) 72 hpf and (**b**) 96 hpf after exposure to different natural water samples. ZnONPs spiked in different water samples resulted in later hatching times in all groups. The significance between each different concentration groups were represented by * sign. (* *p* < 0.05). ZnONPs, zinc oxide nanoparticles; hpf, hours post-fertilization.

**Figure 6 nanomaterials-12-00717-f006:**
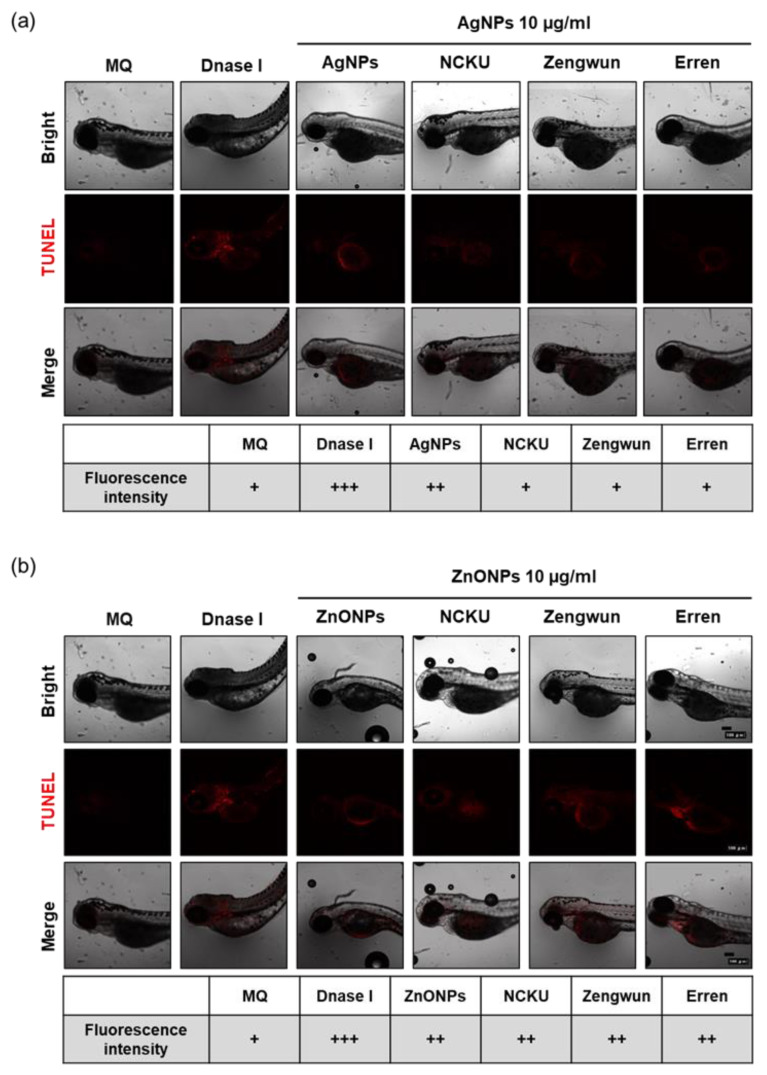
Cytotoxicity of AgNPs and ZnONPs. Zebrafish embryos were exposed to natural water after passing through a 0.45 µm filter and then spiked with NPs. After 72 hpf, zebrafish embryos were subjected to a TUNEL assay. Fluorescence images and intensity of (**a**) AgNPs and (**b**) ZnONPs. Apoptotic cells are represented by red fluorescence signals. The fluorescence intensity was assigned as + (weak), ++ (middle), and +++ (strong). Both AgNPs and ZnONPs spiked in natural water produced increased apoptosis signals in the 10 µg/mL NP groups. AgNPs, silver nanoparticles; ZnONPs, zinc oxide nanoparticles; hpf, hours post-fertilization.

**Figure 7 nanomaterials-12-00717-f007:**
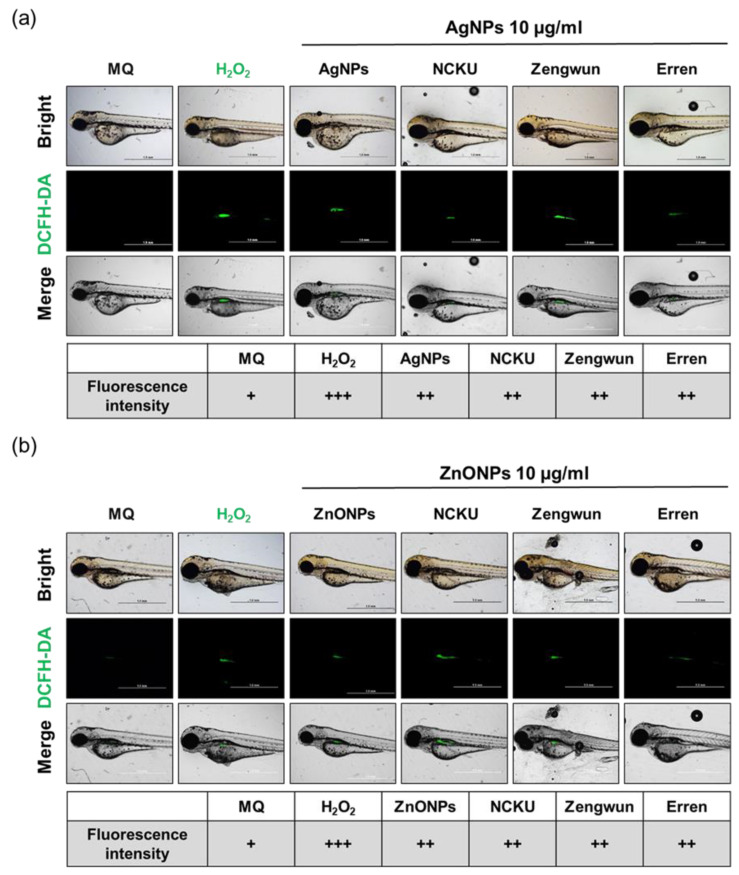
ROS generation induced by AgNPs and ZnONPs. Zebrafish embryos were exposed to natural water after passing through a 0.45 µm filter and then spiked with (**a**) AgNPs and (**b**) ZnONPs. After 72 hpf, zebrafish embryos were stained with DCFH-DA (green). The fluorescence intensity was assigned as + (weak), ++ (middle), and +++ (strong). The H_2_O_2_-positive control group showed ROS generation. Both AgNPs and ZnONPs spiked in natural water resulted in increased ROS signals. AgNPs, silver nanoparticles; ZnONPs, zinc oxide nanoparticles; DCFH-DA, dichlorodihydrofluorescein diacetate; ROS, reactive oxygen species.

**Figure 8 nanomaterials-12-00717-f008:**
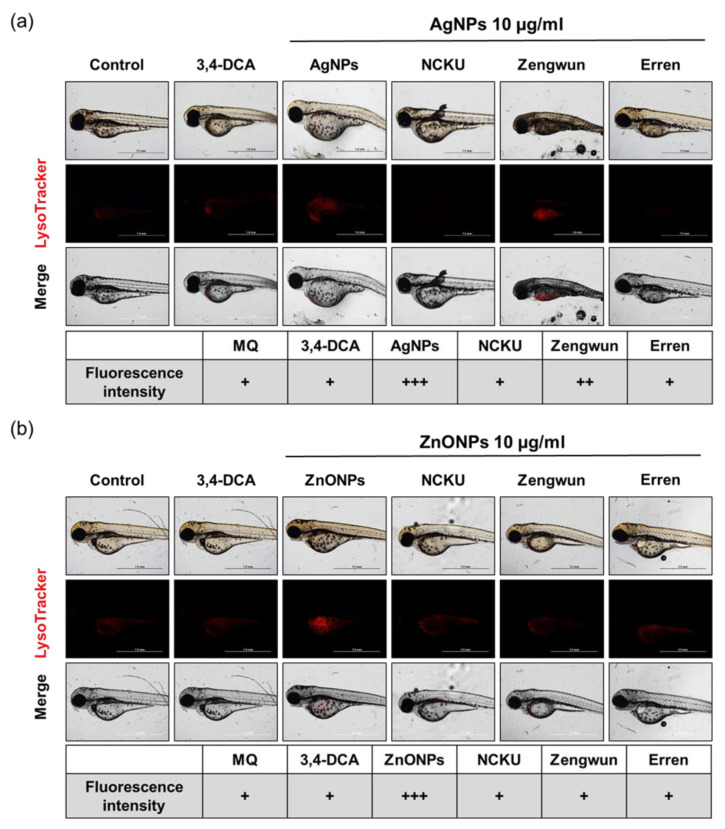
Autophagy induced by AgNPs and ZnONPs. Zebrafish embryos were exposed to natural water after passing through a 0.45 µm filter and then spiked with (**a**) AgNPs and (**b**) ZnONPs. After 72 hpf, zebrafish embryos were stained with LysoTracker RED. The fluorescence intensity was assigned as + (weak), ++ (middle), and +++ (strong). The 10 μg/mL MQ groups exhibited higher autophagy signals than groups treated with NPs spiked in natural water samples. AgNPs, silver nanoparticles; ZnONPs, zinc oxide nanoparticles; 3,4-DCA, 3,4-dichloroaniline; hpf, hours post-fertilization.

**Figure 9 nanomaterials-12-00717-f009:**
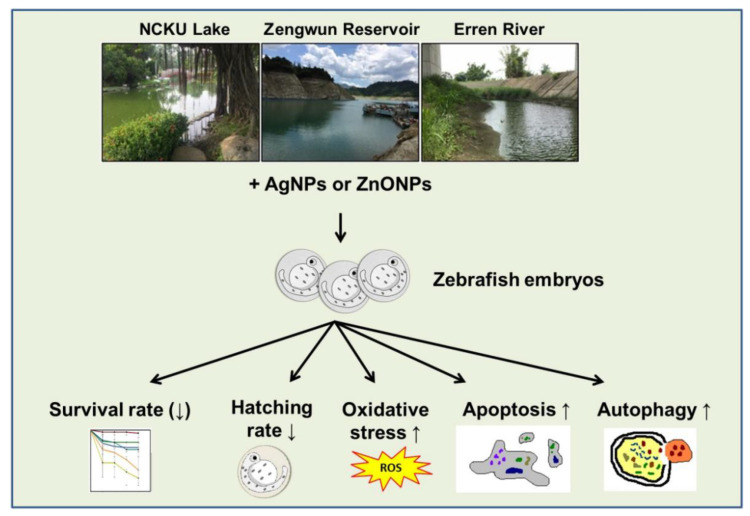
Illustration of the toxic mechanisms of AgNPs and ZnONPs spiked into natural water. The NP suspension was spiked in natural water obtained from NCKU Lake, Zengwun Reservoir, and Erren River and caused lethality and developmental toxicity in embryos. Mechanistically, AgNPs and ZnONPs spiked in natural water induced excessive ROS production, programmed cell death, and overactivated autophagy. AgNPs, silver nanoparticles; ZnONPs, zinc oxide nanoparticles.

## Data Availability

Not applicable.

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
