# Peer review of "Toxic Effects and Mechanisms of Silver and Zinc Oxide Nanoparticles on Zebrafish Embryos in Aquatic Ecosystems"

_nanomaterials, 2022, doi:10.3390/nano12040717_

Round 1

Reviewer 1 Report

This is nice piece of research work and would attract a lot of attention from the scientist and industrialist working in the similar realm. This manuscript could be accepted in nanomaterials after responding to few minor comments as given below:

Page number 3, Line 125-126: ”an NaBH4 solution (100 mM) was slowly and stirred vigorously for 2 hours.” Sentence is not clear and should be rewritten.

Page number 3, Line 125-126: The synthetic AgNP solution should be changed to The synthetized AgNP solution

Page 3, Line 127: it was centrifuged at 7500 g for 30 mins, Is it 7500 g or RPM?

Page 3, Line 128: the solution was centrifuged again at 12500 g for 2 hours, is it 12500 g or RPM?

Reviewer 2 Report

In the manuscript the authors investigated the effects of silver nanoparticles (AgNPs) and zinc oxide nanoparticles (ZnONPs) on zebrafish embryos in aquatic environments. The study is interesting but major revisions are needed for this manuscript :

  • The authors declare that they investigated the toxic effects induced by Ag NPs and ZnO NPs on zebrafish embryos, but what they wrote in the experimental part – subchapter 2.2 Ag/ZnO NPs preparation and characterization – is confusing. It is described the procedure for obtaining amine-coated ZnONPs, more precisely aminopropyl silica-coated ZnO NPs. So, what they investigated forward is not ZnO nanoparticles, but aminopropyl silica-coated ZnO NPs which is different. That is why, in my opinion, the whole manuscript had to be revised and the discussion had to take this into account.
  • Throughout the manuscript, the authors use the expression “Ag/ZnO Nanoparticles” which is also confusing. The readers may understand that Ag Nps are deposited on ZnO nps and it is studied the effect of the resulting material. I think it is necessary to change this way of writing!
  • Line 214-216: the authors said “Element analysis indicated that AgNPs contained silver and ZnONPs were composed of zinc and oxygen, which indicated a very high purity of synthesized Ag/ZnONPs (Figure 1b)”. Analyzing figure 1b, it is clear that both samples contain other elements besides Ag (the first sample contains sulfur and a lot of copper) and Zn and O (the second one contains Silicon. Why nitrogen is not present? It was used APTS to obtain amine-coated ZnONPs). All these elements may have an influence on zebrafish embryos! The discussion should be revised.
  • Polydispersity index for AgNPs has an extremely low value (1.53 x 10-8). I think is a mistake there! Please check!

Round 2

Reviewer 2 Report

The authors responded satisfactorily to all the comments.